# A study on optimization of delayed production mode of iron and steel enterprises based on data mining

Zhiming Shi[1]*, Yisong Li[1], Changxiang Lu[2,3]*

1 School of Economics and Management, Beijing Jiaotong University, Beijing, China, 2 Institute of Public Health & Emergency Management, Taizhou University, Taizhou, Zhejiang, China, 3 Business Colleage, Taizhou University, Taizhou, Zhejiang, China

* 18113047@bjtu.edu.cn (ZS); chauncy.lu@qq.com (CL)

**Data Availability Statement:** All relevant data is available in the following figshare repositories: https://doi.org/10.6084/m9.figshare.21502425 https://doi.org/10.6084/m9.figshare.21502458

## Abstract

Delayed production mode has been adopted by an increasing number of process production enterprises as a method to realize mass customization of multi-products. This paper used the convolutional neural network-long short-term memory artificial neural network algorithm (C-LSTM) in data mining technology to analyze and determine factors that have an impact on delayed production mode in the internal and external production and operation of enterprises. Combined with the actual production situation of iron and steel enterprises, a quantitative model of the delayed production was constructed. Lastly, data from a large iron and steel enterprise with good operation was used to verify the validity of the proposed model and analyze key influencing factors. According to the research, in scenarios of considering PDP alone, considering CODP alone, considering both PDP and CODP, considering PDP and CODP and using data mining technology to model, the matching degree of these methods with the actual situation of the enterprise is 31.8%, 61.4%, 71.6% and 86.6%, respectively. The numerical analysis results of the model based on data mining technology show that in delayed production, when customer service level improves or the delay penalty coefficient increases, the optimal locations of the product differentiation point (PDP) and customer order decoupling point (CODP) move toward the end of production, and the total cost increases gradually. When the difference in production cost or benefit of early delivery between the candidate locations of PDP and CODP is small, optimal locations of PDP and CODP are close to the beginning of the general and dedicated production processes. With an increase of cost difference or early delivery benefit, the optimal locations of PDP and CODP jumped to the end stage of the general and dedicated production processes, and the total cost begins to decrease.

## 1. Introduction

In China, iron and steel enterprises are mainly based on long-process production processes, which are characterized by strong production continuity, product diversification, and complex

https://doi.org/10.6084/m9.figshare.21502467
https://doi.org/10.6084/m9.figshare.21502473.

**Funding:** The authors received no specific funding
for this work.

**Competing interests:** The authors declare that they
have no conflicts of interest.

technological processes. With the development of production technology and the steel market, numerous iron and steel enterprises have begun to introduce delayed production mode from simply using the production-by-storage or production-to-order mode to meet the needs of multi-product and customized production. Enterprises often choose and optimize production models according to production technology, business strategy, and market environment, among others. Effective selection or optimization of production models can encourage enterprises to improve efficiency, reduce costs, and improve customer satisfaction. Initially, the enterprise production mode is mainly divided into the production-to-stock mode (Popp) based on the (s, S) inventory strategy and production-to-order mode [1, 2]. With the development of production technology and market, additional forms of production models have emerged. The delayed production mode can meet the needs of customers for different product production time and different degrees of customization [3, 4]. Delayed production mode is a production mode that appropriately delays production links such as production, assembly, distribution and transportation. Customer order decoupling point (CODP) is the basis for the study of delayed production mode, and the choice of CODP determines the production modes, such as make-to-stock, make-to-order, and design-to-order [5].

Quantitative research on delayed production mode has mainly considered such influencing factors as production time, service level, and total cost or benefit; and has analyzed the locationing of a single CODP or product differentiation point (PDP) by establishing a model or framework. CODP or PDP location generally requires balancing the relationship between market and production [6] or considering the impact of total cost and benefits with the goal of maximizing customer service levels [7]. Some studies have investigated the situation of multiple CODPs or PDPs according to the characteristics of different application scenarios of delayed production mode [8]. Different products have different production processes [9] that are multi-layered [10], and the product series includes different types [11], which all need to consider multiple CODP or PDP location issues. In addition, semi-finished product inventory corresponding to CODP or PDP is important for delayed production control, including the impact of inventory volume on delayed production control [12, 13] and the impact of inventory capacity constraints [14, 15] on delayed production control. According to the literature on delayed production mode and actual situation of delayed production mode in iron and steel enterprises, optimization of delayed production mode in the process of production enterprises should simultaneously consider the content of delayed production control and inventory management.

The goal of using delayed production in the steel industry is to achieve continuous, multi-product, and customized production. CODP is located in the dedicated production process, which reflects the starting point of customer order fulfillment and is also the location for storing dedicated semi-finished product inventory. Evidently, reasonable CODP location can meet product customization goals. PDP is located in the general production process, which is the starting point for the differences in the general production process of various products and is likewise the location for storing the general semi-finished product inventory. Hence, reasonable PDP location can adapt to the production needs of multi-products with general processes. In addition, continuous production will inevitably generate various semi-finished product inventory, and effective control of its inventory distribution can reduce production costs and improve the on-time fulfillment rate of customer orders. Therefore, to use the delayed production mode in a typical multi-product and process production enterprises (e.g., iron and steel enterprise), multiple CODP, multiple PDP, and semi-finished product inventory distribution must be considered simultaneously.

Related influencing factors should be considered when studying the optimization of delayed production mode. Some scholars have studied individual influencing factors, including

delivery time [9] and overall profit [16]. Moreover, most scholars have analyzed the cross-influence of multiple factors on delayed production patterns, such as cost and lead time [17]; cost and benefit [18]; lead time, inventory capacity, and total cost [14] and on-time order fulfillment rate, production cost, and inventory cost [19]. Delayed production in iron and steel enterprises is affected by such factors as internal costs, profits, and delivery dates, and also by various external factors. This paper used data mining technology to analyze and determine the factors that have an impact on delayed production in the internal and external production and operation of enterprises.

With the rapid development of information technology in recent years, the rapid accumulation of various data has also promoted the progress of data mining technology, making it widely used in medicine [20], education [21, 22], management, and manufacturing [23, 24]. In particular, the artificial neural network algorithm in data mining technology is a type of data mining technology inspired by biology, has strong nonlinear expression ability, and is mainly used for prediction research [25, 26] and regression analysis [27, 28]. The current research involves regression and predictive analyses in data mining. Given that there are business relationships and time sequences between the data, the convolutional neural network -long short-term memory (C-LSTM) artificial neural network algorithm is considerably effective because it combines the characteristics of two traditional neural network models (i.e., CNN and LSTM), and can effectively deal with the logical characteristics of space, business, and time existing in data. Zhou et al. [29] used the C-LSTM artificial network algorithm to analyze text data with related expressions and time connections. Kim et al. [30] used the C-LSTM artificial network algorithm to analyze time-continuous business data.

The main innovations of this paper are as follows. First, for continuous production of iron and steel enterprises, combined with the characteristics of time sequences and business relationships of enterprise data, the C-LSTM artificial neural network algorithm in data mining technology is targeted to analyze the impact degree and impact mode of various factors inside and outside the enterprise on delayed production. Second, based on data mining results and the characteristics of delayed production in iron and steel enterprises, a quantitative model of delayed production is constructed and the impact of related factors on delayed production patterns is analyzed based on the proposed model and enterprise data.

The remainder of this paper is organized as follows. Section 2 presents the literature review. Section 3 introduces the data mining tasks, results, and method comparisons. Section 4 builds a quantitative model of delayed production mode based on data mining results and the characteristics of delayed production in iron and steel enterprises. Section 5 analyzes and discusses the important influencing factors in the proposed model. Lastly, Section 6 summarizes the research content of this paper.

## 2. Literature review

The literature related to this research includes three aspects: artificial neural network algorithm, influencing factors of delayed production mode, and delayed production mode in process production enterprises.

### 2.1 Data mining technology and C-LSTM artificial neural network algorithm

According to literature research, data mining technology is widely used in the fields of medicine, education, management, and manufacturing. With the advancement and development of intelligent manufacturing, an increasing number of applications and research on data mining in manufacturing has been observed. Some scholars have analyzed the technological

development and application of data mining from an industrial perspective [23]. Production requirements and data characteristics to improve data development technology [24] focus on quality management [31], manufacturing process optimization [32], and other aspects of optimization research. The focus of this paper is to use data mining technology to optimize the production of enterprises, and the main technology used is the artificial neural network algorithm. The artificial neural network algorithm in data mining technology has strong nonlinear expression ability, but the effect of processing data with time sequences and business relationships is general. Relevant scholars have combined the characteristics of convolutional neural network (CNN) and long and short-term memory (LSTM) models to form the C-LSTM artificial neural network algorithm with different characteristics, which can simultaneously process data with business, spatial, and temporal relationships. Zhang [33] and Zhou et al. [29] used the C-LSTM artificial neural network algorithm to process data with business relationships and temporal order to achieve effective classification. Kim et al. [30] used the C-LSTM artificial neural network algorithm to process data with spatial relationship and temporal characteristics and to monitor abnormal data on the Internet. According to research requirements, this paper used the C-LSTM artificial neural network to process data with business relationships and time sequences. In the front section, a one-dimensional convolution layer was used to extract the logical relationship of production business. Moreover, the internal coupling characteristics of each signal were extracted by reducing sampling and filtering. In the back section, a long-time short memory layer was used to extract the temporal logic characteristics to realize the function of immediately and effectively extracting data characteristics.

## 3. Data mining to analyze the influencing factors of delayed production

This paper used data mining technology to determine factors that strongly influence delayed production inside and outside enterprises. In addition, this study analyzed the influence mode of these factors.

### 3.1 Task description of data mining

This study used a large-scale iron and steel enterprise with good operating conditions and using the delayed production mode as basis in collecting and sorting out production and operation data and delayed production control data of the enterprise within five years at weekly intervals. Moreover, the current research utilized the C-LSTM artificial neural network algorithm of data mining technology to conduct regression analysis to determine the factors and influence modes that strongly impact delayed production control in production and operation. Thereafter, the Pearson correlation coefficient between the factors that strongly impact delayed production was calculated, the correlation between these factors was determined, and the potential causal relationship between the factors was excluded from the impact of the data mining results.

The production and operation data and delayed production control data of the enterprise came from the iron and steel enterprise information system and social information platform, including production, customer management, energy, sales, procurement, and measurement systems; and e-commerce and steel information platforms. Among them, the production and operation data factors can be divided into three aspects: production, operation, and market (see Fig 1). Delayed production control data came from three products: hot-rolled, chilled, and galvanized. The three products are typical low-, medium-, and high-grade products in iron and steel enterprises. Delayed production control data include CODP location, semi-finished product inventory corresponding to CODP, PDP location, and semi-finished product

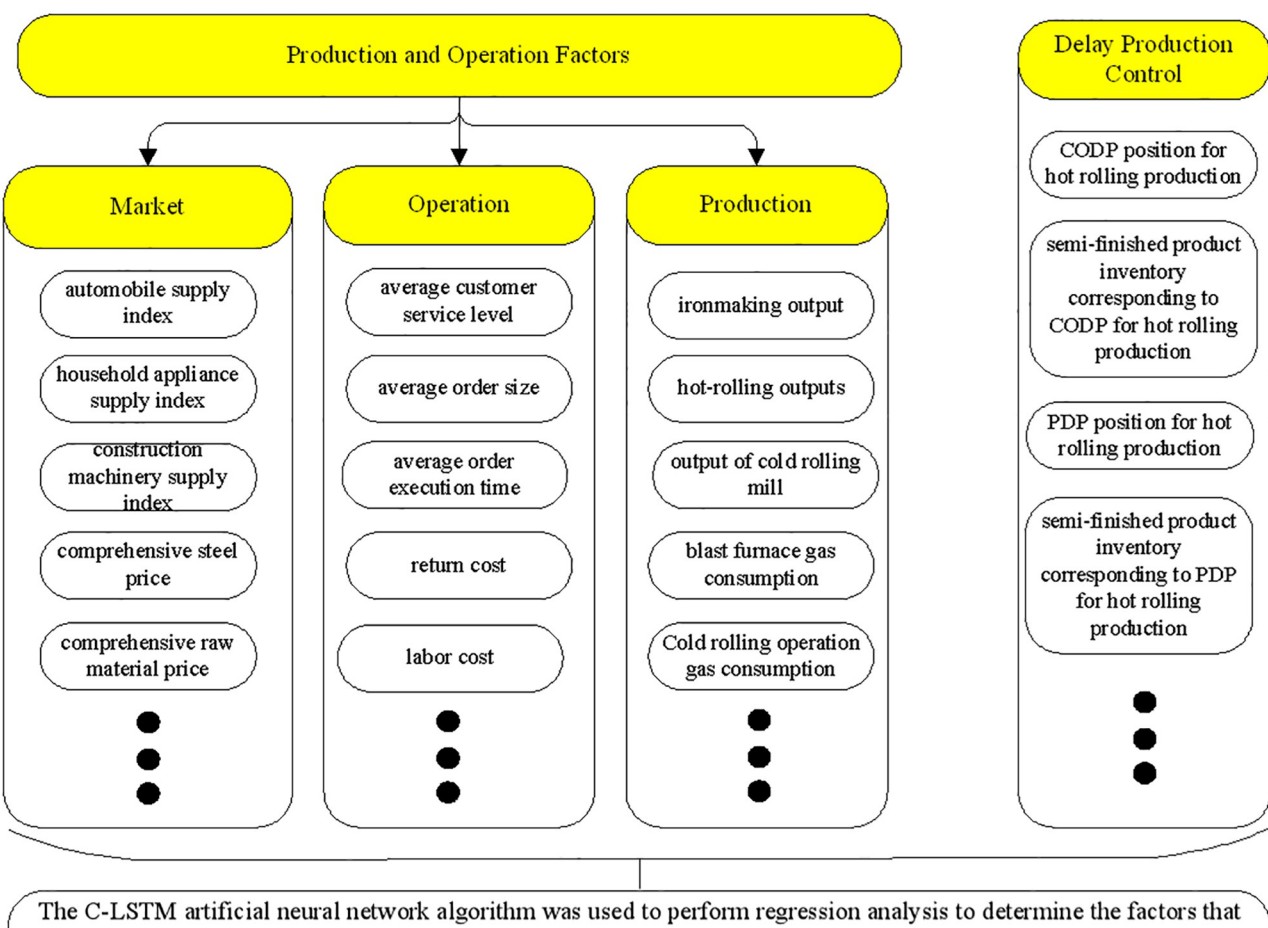

**Fig 1. Data mining tasks and implementation steps.**

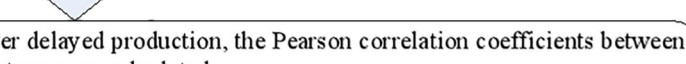

inventory corresponding to PDP. Data mining tasks, methods used, main data, and implementation steps are shown in Fig 1.

Fig 1 shows that data mining tasks and implementation steps are as follows. First, the C-LSTM artificial neural network algorithm was used to perform regression analysis to determine the factors that had strong control over delayed production in production and operation. Second, for the determined factors that had strong control over delayed production, the Pearson correlation coefficients between the factors were calculated. Lastly, combined with the production and operation situation and research needs, excluding unreasonable factors, this study determined the production and operation factors that should be considered and have a strong impact on delayed production control.

In this paper, combined with the research needs, C-LSTM artificial neural network is used to process the data with business relationships and time sequence. In the front section, a dimension convolution layer is used to extract the production business logic relations, and the internal coupling characteristics of each signal are extracted by reducing sampling and filtering. In the back end, a long time short memory layer is used to extract the temporal logic characteristics, thus realizing the function of fast and effective extraction of data characteristics.

The main task of data mining in this paper is to use the technology of C-LSTM artificial neural network algorithm, which is to process the data with business relationships and time sequences. In the front part, a one-dimensional convolution layer is used to extract the business relationships, and the intrinsic coupling characteristics of each signal are extracted by reducing sampling and filtering. In the back part, long-time and short-memory layer is used to extract the time sequences, thus realizing the function of fast and effective data extraction. The analysis results are shown as follows.

## 3.2 Analysis of data mining

This paper used the C-LSTM artificial neural network algorithm, PyCharm as integrated development environment, and Python 3.9 for programming, and analyzed production and operation data and delay control data.

All data are from a large iron and steel enterprise with good operation or information platform. The data collection scope is 5 years, and the time unit is weekly, including 17 market factors, 8 business factors, 11 production factors, and 12 delayed production control data. According to the confidentiality requirements of the enterprise, data were encrypted without affecting the analysis results. The data used for data mining analysis is "The enterprise production and operation data and delayed production control data". These data are shared on the public repository ("Figshare"). You can find the sharing link in the "Data availability statement". The results of the analysis are as follows.

**3.2.1 Relationship between production and operation factors and delayed production control.** This paper analyzed the relationship between production and operation factors and delayed production control, including hot-rolled, chilled, and galvanized products. The influence degree is expressed by the root mean square error (RMSE). The specific analysis results are as follows.

(1) Hot-rolled products

For the hot-rolled products, the influence of different production and operation factors on the delayed production control is shown in Fig 2.

Fig 2 shows that in the production of hot-rolled products, production and operation factors that strongly impact delayed production control include hot-rolled output, building materials supply index, construction machinery supply index, and ironmaking output. Such information as building materials supply index and construction machinery supply index came from the "My Steel Network platform", which is used to display the supply of steel products to other industries by the steel industry within a certain period.

(2) Chilled products and galvanized products

According to data mining results, in the delayed production of chilled and hard products, production and operation factors that strongly impact he delayed production control include the automobile supply, construction machinery supply, comprehensive steel price, and building material supply index. In the delayed production of galvanized products, production and

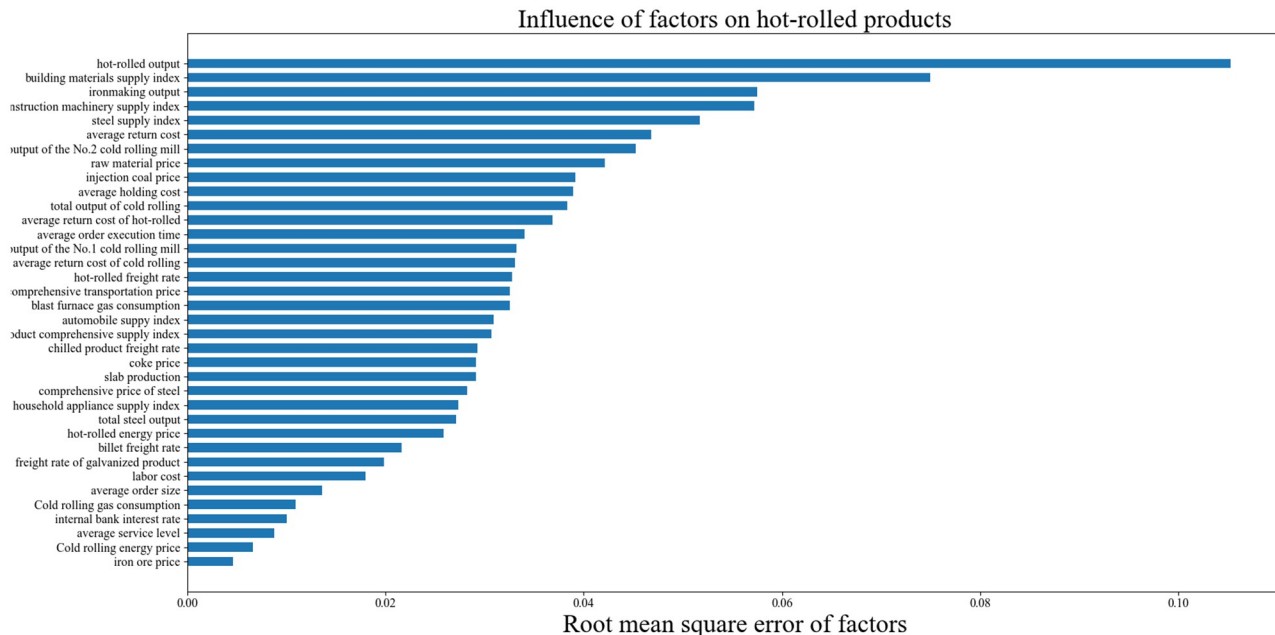

**Fig 2. Influence of factors on delayed production mode of hot-rolled products.**

operation factors that strongly impact the delayed production control include automobile supply index, construction machinery supply index, and comprehensive steel price.

(3) Summary of analysis results of the three products

The data mining analysis results of the impact of delayed production control are shown in Table 1.

Table 1 shows that in the production of three products, the factors that strongly impact the delayed production control are as follows.

1. Auto, construction machinery, and building material supply index and comprehensive steel price in the market strongly impact delayed production control. Investigation and analysis showed that the higher the supply index of each industry, the greater the demand for steel in the industry, often causing the price of steel to increase. At this time, iron and steel enterprises will inevitably accelerate production, thereby affecting the control of enterprises to delay production.

**Table 1. Data mining analysis results of the impact of production and operation factors on delayed production control.**

| delayed production Control of different products | market | | | | operation | production | |
| --- | --- | --- | --- | --- | --- | --- | --- |
| production and operation factors | automobile supply index | construction machinery supply index | building material supply index | comprehensive steel price | return costs | ironmaking outputs | hot-rolled outputs |
| **hot-rolled products** | | strong correlation | strong correlation | | strong correlation | strong correlation | strong correlation |
| **chilled products** | strong correlation | strong correlation | strong correlation | strong correlation | | | |
| **galvanized products** | strong correlation | strong correlation | | strong correlation | | | |

2. Average return cost of operation has a strong impact on delayed production control. Return costs are various costs incurred when semi-finished products in production are delayed to enter the production process again. Return costs include the transportation cost of semi-finished products and energy consumption cost when semi-finished products enter the production process again.

3. Ironmaking and hot-rolled outputs in production also have a strong impact on delayed production control. Investigation and analysis indicated that changes in ironmaking and hot-rolled outputs affect the storage of semi-finished products in delayed production control but have minimal impact on delayed production control. However, the impact of ironmaking and hot-rolled outputs will not be considered in the follow-up.

Results of data mining and the production and operation of enterprises indicated that the average return cost, comprehensive steel price, automobile supply index, building material supply index, and construction machinery supply index have a strong impact on delayed production control.

**3.2.2 Correlation analysis between factors with strong influence.** Correlation analysis was conducted for the determined factors with a strong influence on delayed production control. The Pearson correlation coefficient analysis method was used, and the analysis results included positive and inverse correlations, representing positive promotion and negative influence effects, respectively (Table 2).

Table 2 shows the correlation analysis results of various influencing factors. Among them, building materials supply index was significantly positively correlated with construction machinery supply index, comprehensive steel price was significantly positively correlated with building materials and construction machinery supply index, and the correlation between other influencing factors was not strong. Combined with the analysis results in Table 2 and the actual production and operation situation of the enterprise, this paper conducts a more in-depth analysis, and the analysis results are as follows:

1. Building materials supply index was significantly positively correlated with construction machinery supply index. That is, the two steel products have a certain supply homogeneity and production linkage. This positive correlation has no effect on data mining results. Therefore, the building materials and construction machinery supply index have a strong impact on delayed production.

2. Comprehensive price of steel was significantly positively correlated with the supply index of building materials and construction machinery. That is, comprehensive price of steel was affected by the supply index of building materials and construction machinery. This impact is a trend of market operation coordination and not a direct causal relationship, and has no impact on data mining results. Therefore, comprehensive price of steel has a strong impact on delayed production.

**Table 2. Correlation analysis results between various influencing factors.**

|  | return costs | comprehensive steel price | automobile supply index | construction machinery index | building material supply index |
|---|---|---|---|---|---|
| **return costs** | 1.00 | 0.24 | 0.14 | -0.11 | -0.26 |
| **comprehensive steel price** |  | 1.00 | 0.05 | 0.55 | 0.54 |
| **automobile supply index** |  |  | 1.00 | 0.33 | 0.20 |
| **construction machinery index** |  |  |  | 1.00 | 0.76 |
| **building material supply index** |  |  |  |  | 1.00 |

Combined with the results of data mining, correlation analysis, and the production and operation of enterprises, the comprehensive research showed that the average return cost, comprehensive steel price, automobile supply index, building material supply index, and construction machinery supply index have a strong impact on delayed production.

**3.2.3 Analysis of the influence of related factors on delayed production control.** The preceding research determined the factors with strong influence on delayed production control. Combined with the three products involved in this study, the influence mode and degree of the relevant factors on delayed production control were further analyzed.

Given that chilled products are middle-grade products in iron and steel enterprises, they can be used as semi-finished or finished products. Accordingly, the influencing factors of delayed production control of chilled products should be analyzed. Data mining results showed that the factors with a strong influence on the delayed production control of chilled products include automobile supply index, construction machinery supply index, comprehensive steel price, and building material supply index. The ways and degrees of their influence are as follows.

(1) Analysis of the influence of automobile supply index on the delayed production control of chilled products

Fig 3 shows that with an increase in the automobile supply index, the index constantly promoted the CODP location to move to the right. The index initially promoted the PDP location to the right and to the left thereafter, but the influence of the left movement was minimal. The index has minimal effect on the proportion of CODP semi-finished products inventory.

(2) Analysis of the influence of construction machinery supply index on delayed production control of chilled products

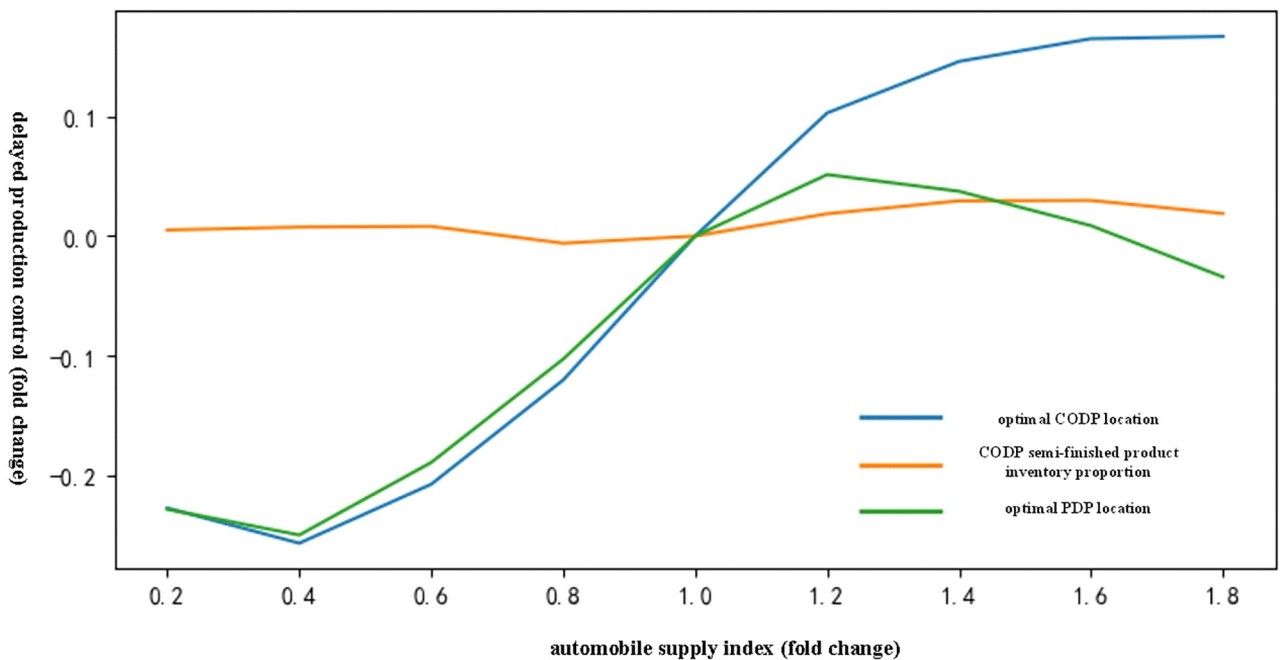

**Fig 3. Influence trend of automobile supply index on delayed production mode of chilled products.**

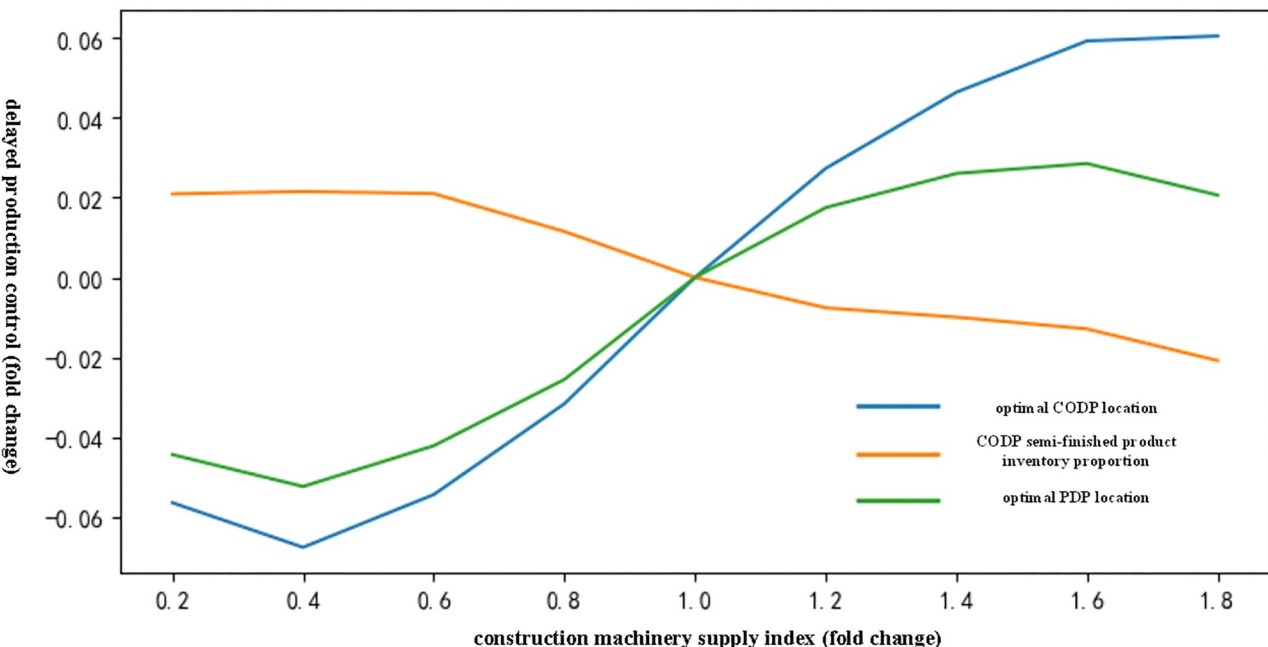

**Fig 4. Influence trend of construction machinery supply index on delayed production mode of chilled products.**

Fig 4 shows that with an increase in the construction machinery supply index, the index promoted the locations of CODP and PDP to move to the right and promoted the reduction of inventory ratio of the CODP semi-finished products. However, the impact was minimal.

(3) Analysis of the influence of comprehensive steel prices on delayed production control of chilled products

Fig 5 shows that the increase in the comprehensive price of steel pushed the location of CODP to move to the right. The comprehensive price of steel initially promoted the location of PDP to the right and slightly to the left thereafter, accompanied by an increase in the proportion of CODP semi-finished products inventory.

(4) Analysis of the influence of building materials supply index on delayed production control of chilled products

Fig 6 shows that with an increase in the building materials supply index, the CODP location gradually moved to the right. The index initially promoted the PDP location to move to the right and move slightly to the left thereafter, accompanied by an increase in the proportion of CODP semi-finished products inventory.

The analysis process and results of hot-rolled products and galvanized products are similar to those of chilled products and will no longer be described in detail.

### 3.3 Comparison of data mining algorithms

For data mining task, this paper used the C-LSTM artificial neural network algorithm, random forest algorithm and convolution neural network(CNN) algorithm for calculation. The minimum regression error, maximum regression error and average regression error of these algorithms are shown in Table 3.

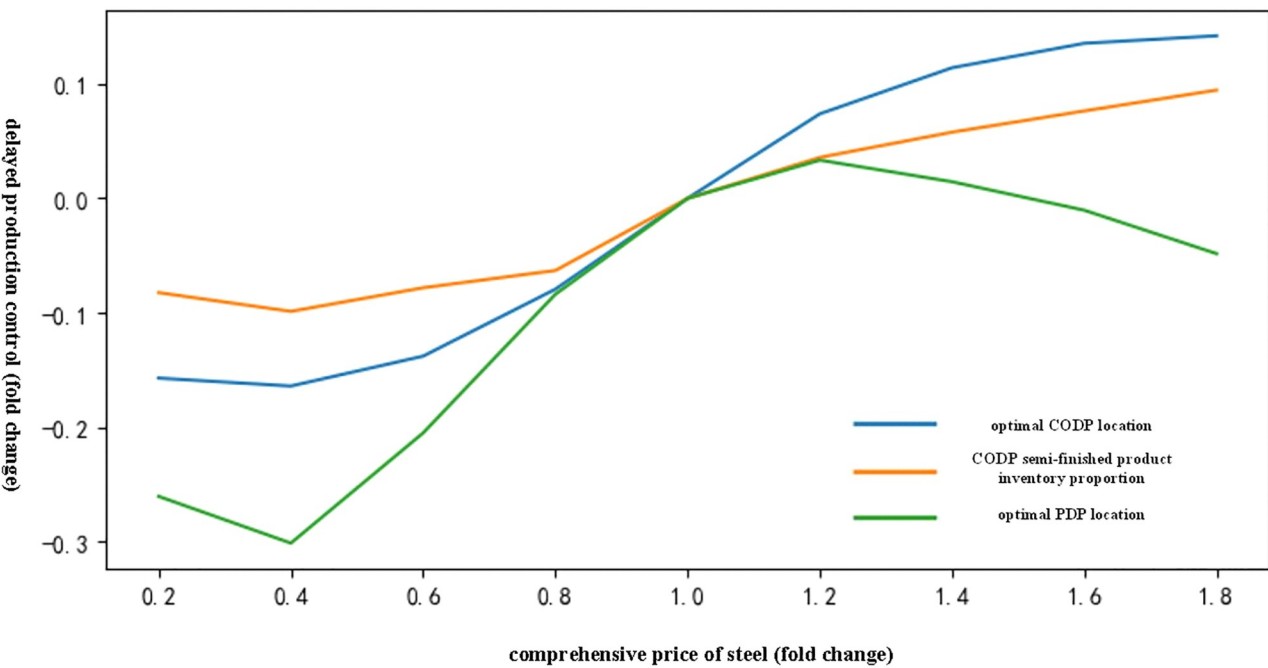

**Fig 5. Influence trend of the comprehensive price of steel on delayed production mode of chilled products.**

The regression errors of C-LSTM artificial neural network algorithm and random forest algorithm are analyzed in detail, and the results are shown in Fig 7.

As shown in Fig 7, the horizontal axis (x-axis) represents the regression error of hot-rolled products, the longitude axis (y-axis) represents the regression error of chilled products, and

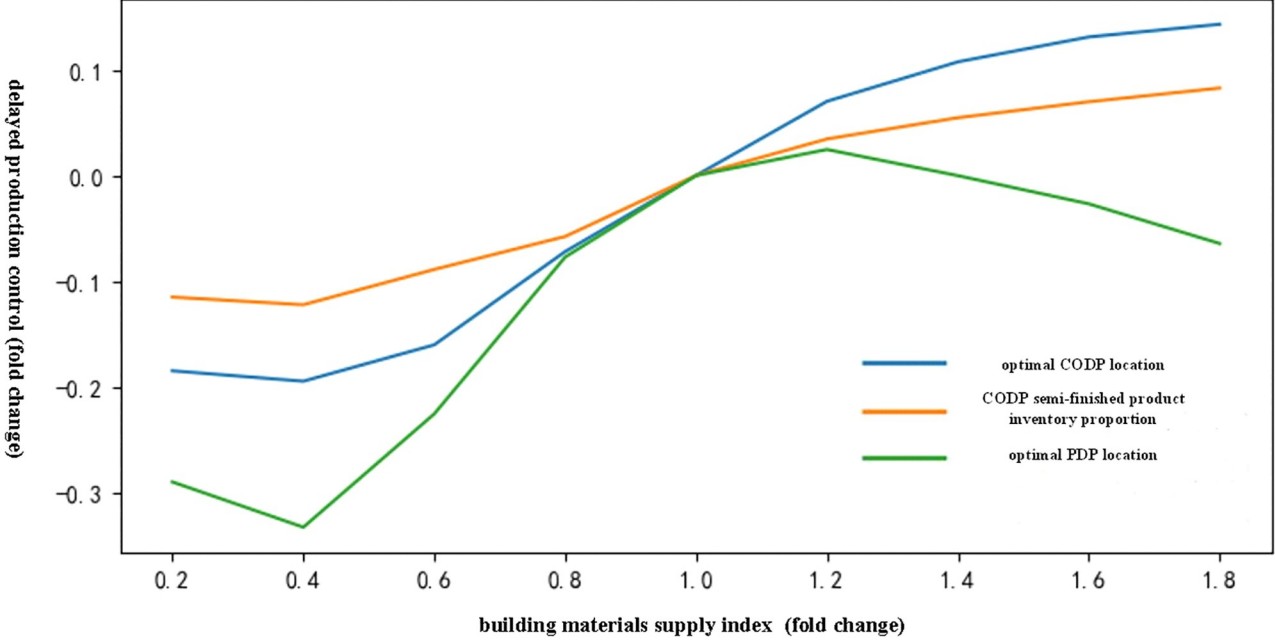

**Fig 6. Influence trend of the building materials supply index on delayed production mode of chilled products.**

**Table 3. Regression error comparison of different algorithms.**

|  | Hot-Rolled Products | | | Chilled Products | | | Galvanized Products | | |
|---|---|---|---|---|---|---|---|---|---|
|  | Max | Min | Average | Max | Min | Average | Max | Min | Average |
| C-LSTM | 0.45 | 0.008 | 0.1 | 0.34 | 0.011 | 0.085 | 0.34 | 0.009 | 0.075 |
| Random Forest | 0.35 | 0.039 | 0.13 | 0.33 | 0.022 | 0.097 | 0.26 | 0.021 | 0.076 |
| CNN | 0.43 | 0.041 | 0.15 | 0.32 | 0.025 | 0.103 | 0.33 | 0.025 | 0.085 |

the vertical axis (z-axis) represents the regression error of galvanized products. The blue dot indicates the result of using the C-LSTM artificial neural network algorithm, and the red dot indicates the result of using the random forest algorithm. The blue dots are concentrated and closer to the origin, while the red dots are scattered and far away from the origin. Therefore, the comparative analysis of the algorithms indicated that the regression error of the C-LSTM artificial neural network algorithm was smaller, and thus more effective.

## 3.4 Influencing factors of delayed production mode

According to the relevant literature, the research on delayed production mode has often considered such factors as cost, profit, delivery time, inventory, and risk. A few studies have also considered the influence of unilateral factors on delayed production mode. Sun et al. [9] factored in the influence of delivery time, used the bill of materials as basis in studying the locationing of multiple decoupling points in the supply chain, and constructed a multi-decoupling point of mathematical model. To maximize profit. Most of the literature has considered the impact of multiple factors on delayed production patterns. Cuypere et al. [12] considered the delivery time and total cost, and used the Markov decision model to study the "decoupled inventory" problem. Kuthambalayan and Bera [34] considered the impact of delivery time and

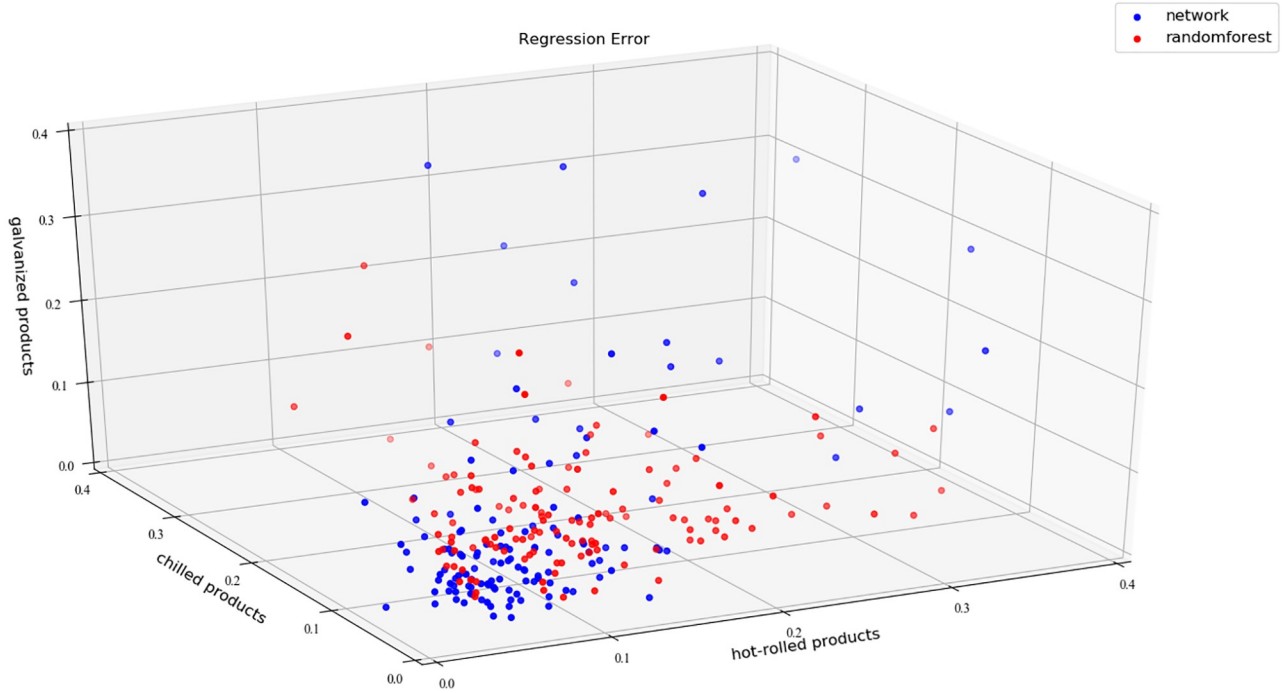

**Fig 7. Regression error comparison of C-LSTM artificial neural network algorithm and random forest algorithm.**

total profit and used a two-stage model to study the inventory of semi-finished products with delayed production. Lee et al. [18] considered the impact of costs and benefits and built a model to optimize delayed production, with the goal of achieving product diversity and improving customer service levels. Ji et al. [14] considered the impact of delivery time, inventory capacity, and total cost; and conducted research on CODP in the process industry. Sharda [19] considered the impact of on-time order fulfillment rate, production cost, and inventory cost; and took a chemical factory as research object to build a quantitative model to study the optimization of MTS and delayed production mode.

For delayed production mode, the influencing factors considered in the relevant literature have mainly originated from within enterprises, and minimal consideration has been given to their external influencing factors. However, production mode has often been affected by the external production and operation factors of enterprises, the impact of which is significant. This paper is convinced that the factors affecting delayed production mode from inside and outside enterprises should be comprehensively analyzed from the three aspects of market, operation, and production.

### 3.5 Research on delayed production mode in process production enterprises

According to the relevant literature, the focus of the research on delayed production mode in process production enterprises is the research on CODP location. Some studies have also considered the content of inventory management. For the food industry with process production, Van Donk [6] optimized the production mode combined with such influencing factors as production, market, product, and inventory. He was convinced that CODP location is the focus of delaying the optimization of production mode, and the optional location of CODP corresponds to inventory at all levels. Zandieh et al. [17] combined the characteristics of process industries with multiple products and multiple inventories and studied the location of the best CODP. They were convinced that inventory management is as important as the content of production control, and inventory capacity constraints and storage characteristics also have an impact on delayed production control. Sharda [19] considered the characteristics of production continuity, product diversification, and temporary storage for a chemical company with process production, combined with the delayed production and production-by-storage modes used by this enterprise. They were convinced that in the optimization of the production mode, storage location of the semi-finished product inventory and influence of the storage container should be considered. For the delayed production mode in process production enterprises, particularly the production of multiple products and different processes, some studies have also considered the impact of PDP in addition to CODP and inventory management. Vanteddu et al. [35] and AlGeddawy et al. [36] studied the location problem of PDP by building models in different scenarios, but mainly for the delayed production of discrete production enterprises.

According to the literature research, the research of data mining includes definitions, technologies, algorithms, tools, fields, scenarios, applications, etc. In the application of data mining, it is concentrated in the fields of medicine, education, management and manufacturing, but the number of the latest literature on the application of manufacturing is few.

The research on the delayed production mode in the literature focused on the supply chain or discrete enterprises. Generally, PDP and CODP were considered separately. A small number of literature considered PDP and CODP at the same time. In addition, the focus of literature research includes CODP location, PDP location, CODP location and PDP location, CODP and inventory managemet. According to the actual situation of delayed production

mode in continuous production enterprises and literature research, this paper proved that CODP location, PDP location, and various semi-finished product inventory distribution should be considered simultaneously to achieve effective control of delayed production.

## 4. Construction and solution of delayed production model

### 4.1 Question description

Delayed production process in iron and steel enterprises mainly includes two steps. The first step is called "production preparation stage". That is, the initial production capacity is converted into different semi-finished products inventory, including general and special semi-finished product inventories corresponding to PDP and CODP, respectively. The second step is "order fulfillment stage", which uses the general and dedicated semi-finished product inventories to complete the production of the product according to customers' order requirements.

The specific production process of delayed production is shown in Fig 8. In the general production process of delayed production, one location for each product category was selected as PDP among multiple PDP candidate locations, and a certain amount of general semi-finished product inventory was stored. In the dedicated production process of delayed production, one location for each product was selected as CODP among multiple CODP candidate locations, and a certain amount of dedicated semi-finished product inventory was stored.

Comprehensive analysis showed that delayed production control includes four aspects: PDP location, semi-finished product inventory corresponding to PDP, CODP location, and semi-finished product inventory corresponding to CODP. Among the four aspects, PDP in the general production process indicates the starting point of differentiation in the production of different products in a certain product category. Semi-finished product inventory corresponding to PDP belongs to the general semi-finished product inventory that can be used for the production of various products in this product category. CODP in the dedicated production process represents the starting point for the enterprise to start production according to customer order requirements. Semi-finished product inventory corresponding to CODP is a

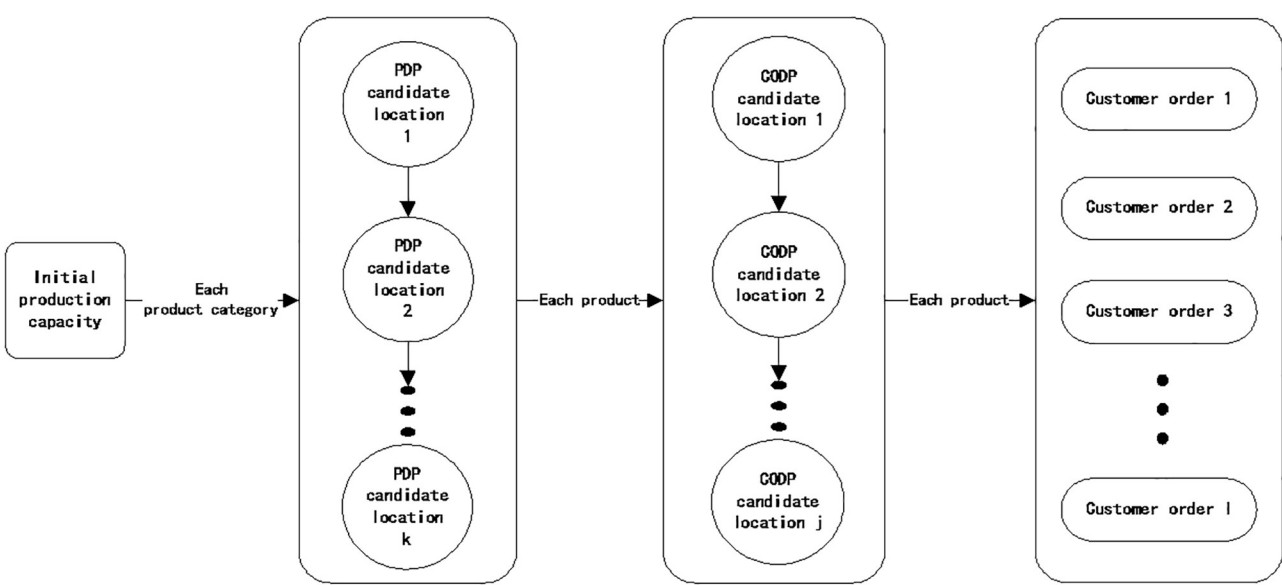

**Fig 8. Process of delayed production mode for steel companies.**

special semi-finished product inventory that can only be used for the production of a certain product.

According to production and operation management, each CODP and PDP candidate location corresponds to different unit costs, and customer order requirements of each product are also different. Therefore, selecting different candidate locations as CODP or PDP and storing different quantities of semi-finished products inventory at this location will affect the total cost and service level. By constructing a research model, this paper quantitatively studied the optimal decision to achieve the lowest total cost under the service level.

Iron and steel enterprises are typical continuous production enterprises. Delayed production will inevitably produce redundant semi-finished products inventory. Some of the inventory is converted into the semi-finished products inventory corresponding to CODP, the rest of the inventory is usually converted into the semi-finished products inventory corresponding to PDP. Therefore, the model should consider PDP, CODP and semi-finished products inventory at the same time. However, relevant factors considered in previous models are mainly by production management experience and financial indicators, these factors belong to the internal factors of the enterprise, and the influence of external factors of the enterprise needs to be obtained and verified through data mining. The above data mining has determined that various supply indexes and comprehensive product prices have strong independent effects on delayed production. Therefore, when building a delayed production model, external factors need to be transformed into relevant parameters. Based on this research idea, the model is constructed as follows.

## 4.2 Model construction

This paper established an operations research model in response to the preceding problems raised. Given that the proposed model aims to optimize the production model, this type of research often considers the impact of the total cost, including holding and delay penalty costs [10, 37]. In the actual production process of iron and steel enterprises, when the semi-finished product inventory enters the production process again, high transportation costs and large heating energy consumption will be incurred, which can be converted into return costs and included in the total cost.

To accurately and comprehensively describe the delayed production situation in iron and steel enterprises, additional influencing factors should be considered when constructing a quantitative model. This paper used the data mining method for analysis, and the results showed that various steel supply index and comprehensive steel prices have considerable impact on delayed production control. Further analysis and research showed the following results. On the one hand, the change in various steel supply index reflects the supply and demand of steel products, thereby affecting the production speed and raw material price of the product, as well as the production cost of the product. On the other hand, the supply and demand situation and comprehensive price of steel reflected by the changes of various steel supply index will affect the income of early delivery of products. Therefore, this paper converted various steel supply index and comprehensive steel prices into production costs and early delivery benefits, and incorporated them into the total cost. Accordingly, the total cost considered by the proposed model include the holding, delay penalty, turn-back, and production costs and early delivery benefit.

On the basis of the preceding analysis, the proposed model should consider such factors as holding, delay penalty, return, and production costs; early delivery benefit; and customer service level. The decision-making content of the model includes PDP location, CODP location, and semi-finished product inventory. The goal of the proposed model is to build a quantitative

research model to minimize the total cost under the premise of satisfying a specific customer service level. The specific objective functions and constraints are as follows.

(1) Objective functions

$$MinZ = C = C_h + C_r + C_d + C_p - I_e$$

$C$ represents the total cost, $C_h$ represents the holding cost, $C_r$ represents the return cost, $C_d$ represents the delay penalty cost, $C_p$ represents the production cost, and $I_e$ represents early delivery benefit. The total cost is $C = C_h + C_r + C_d + C_p - I_e$. The objective function is to minimize the total cost.

(2) Constraints

$$R_{pr} \geq R_{sl}$$

$$X^{km} = 0 \text{ or } 1$$

$$Y^{jmn} = 0 \text{ or } 1$$

$$\sum_{k^m=1}^{K^m} X^{km} = 1$$

$$\sum_{j^{mn}=1}^{J^{mn}} Y^{jmn} = 1$$

$$0 \leq Q^{km} \leq 5000$$

$$0 \leq Q^{jmn} \leq 5000$$

$R_{pr}$ represents the average customer order fulfillment rate on time and $R_{sl}$ means the minimum customer service level. Meanwhile, $R_{pr} \geq R_{sl}$ represents that the average on-time fulfillment rate of customer orders of enterprises should be higher than the lowest customer service level.

$X^{km} = 0$ or 1 because $X^{km}$ is a variable of 0–1, indicating whether or not a location (i.e., $k^m$th location) is selected as PDP for a product category (the mth product category); $\sum_{k^m=1}^{K^m} X^{km} = 1$ indicates that a product category has one and only one PDP.

$Y^{jmn} = 0$ or 1 because $Y^{jmn}$ is a variable of 0–1, indicating whether or not a location ($j^{mn}$th location) is selected as CODP for a product ($n^m$th product); $\sum_{j^{mn}=1}^{J^{mn}} Y^{jmn} = 1$ represents that a product category has one and only one CODP.

$Q^{km}$ means the corresponding general semi-finished product inventory when the m-th product category selects the $k^m$ candidate location as PDP; $0 \leq Q^{km} \leq 5000$ indicates that the inventory of semi-finished products corresponding to a certain product category cannot be above 5000.

$Q^{jmn}$ means the corresponding special semi-finished product inventory when the $n^m$-th product selects the $j^{mn}$ candidate location as CODP; $0 \leq Q^{jmn} \leq 5000$ indicates that the semi-finished product inventory corresponding to a certain product CODP cannot be above 5000.

### 4.3 Model solution

This paper is based on the production and operation data of a steel enterprise, considering the privacy requirements and research needs of the enterprise, and adopts the method of simulating the market environment and production environment to solve and analyze. The main data include the following aspects: production cycle is 30 days, initial production capacity is 60,000, inventory capacity of semi-finished products corresponding to PDP and CODP is limited to 5,000, and minimum customer service level is 0.85. Other information includes product category PDP candidate location information, product CODP candidate location information, product order information, etc. These data are encrypted according to the requirements of the enterprise. The data used for model solution include "The costs and production time of the PDP candidate locations of different products", "The costs and production time of the CODP candidate locations of different products" and "The order information of different products". These data are shared on the public repository ("Figshare"). You can find the sharing link in the "Data availability statement".

The model constructed in this paper is nonlinear mixed integer programming model, which is solved by using a precise algorithm and realized by lingo software. The optimal decision is as follows.

$$\{PDP : [4000, 0, 0, 0, 0], CODP : [[0, 0, 0, 0, 0, 4000], [0, 0, 0, 0, 0, 4000], [1000, 0, 0, 0, 0, 0]]\}$$

$$\{PDP : [5000, 0, 0, 0, 0], CODP : [[4000, 0, 0, 0, 0, 0], [0, 0, 0, 0, 0, 4000], [0, 0, 0, 0, 0, 4000]]\}$$

$$\{PDP : [5000, 0, 0, 0, 0], CODP : [[0, 0, 0, 0, 0, 4000], [0, 0, 0, 0, 0, 4000], [0, 0, 0, 0, 0, 4000]]\}$$

Taking the first row of the operation result as an example, it means that for the first product category, the optimal location of PDP is 1 and the optimal inventory quantity of the general semi-finished product corresponding to PDP is 4000. For the first product in the first product category, the optimal CODP location is 6 and the optimal inventory of CODP corresponding to special semi-finished products is 4000. For the second product in the first product category, the optimal CODP location is 6 and the optimal inventory of special semi-finished products corresponding to CODP is 4000. For the third product in the first product category, the optimal location of CODP is 1 and the optimal inventory of CODP corresponding to special semi-finished products is 1000. The best location of other product categories and products and the inventory of semi-finished products are similar to the above analysis.

The above results are models constructed based on data mining results. This paper constructed models for considering PDP alone, CODP alone, considering both PDP and CODP. The research results show that, considering PDP alone, CODP alone, considering both PDP and CODP, considering PDP and CODP and using data mining technology to model, the matching degrees of these methods with the actual situation of the enterprise are 31.8%, 61.4%, 71.6% and 86.6%, respectively. So, the model constructed based on data mining technology in this paper can more accurately describe the situations of delayed production in iron and steel enterprises.

## 5. Discussion

In the preceding model solution, the relevant parameter values are fixed. However, in actual production, parameters change at any time, thereby affecting the decision-making content. According to the actual situation of the enterprise, this paper conducts a simulation numerical analysis, and discusses the influence of the changes of several important factors on optimal decision-making and total cost.

The main parameters of the model constructed in this paper include: holding cost, return cost, delay penalty cost, production cost, early delivery benefit, customer service level, etc. The decision-making content includes: CODP location, PDP location, CODP corresponding semi-finished product inventory, PDP corresponding semi-finished product inventory.

For the management of delayed production in iron and steel enterprises, customer service level, early delivery benefit, and production delay penalty cost are more important for enterprise operation. Compared with holding cost and return cost, production cost changes more frequently and is more important to the production process of enterprises. Therefore, this paper analyzes the customer service level, delay penalty coefficient, early delivery benefit coefficient and production cost, and studies the impact of their numerical changes on the delayed production control.

## 5.1 Influence of customer service level

By fixing other parameters and changing customer service levels, analyze the impact of customer service level change on CODP optimal location, PDP optimal location, and total cost was analyzed.

(1) The impact of changes in customer service level on the optimal CODP location and total cost

The optimal location and total cost of CODP for hot-rolled, galvanized, and chilled products are shown in Fig 9.

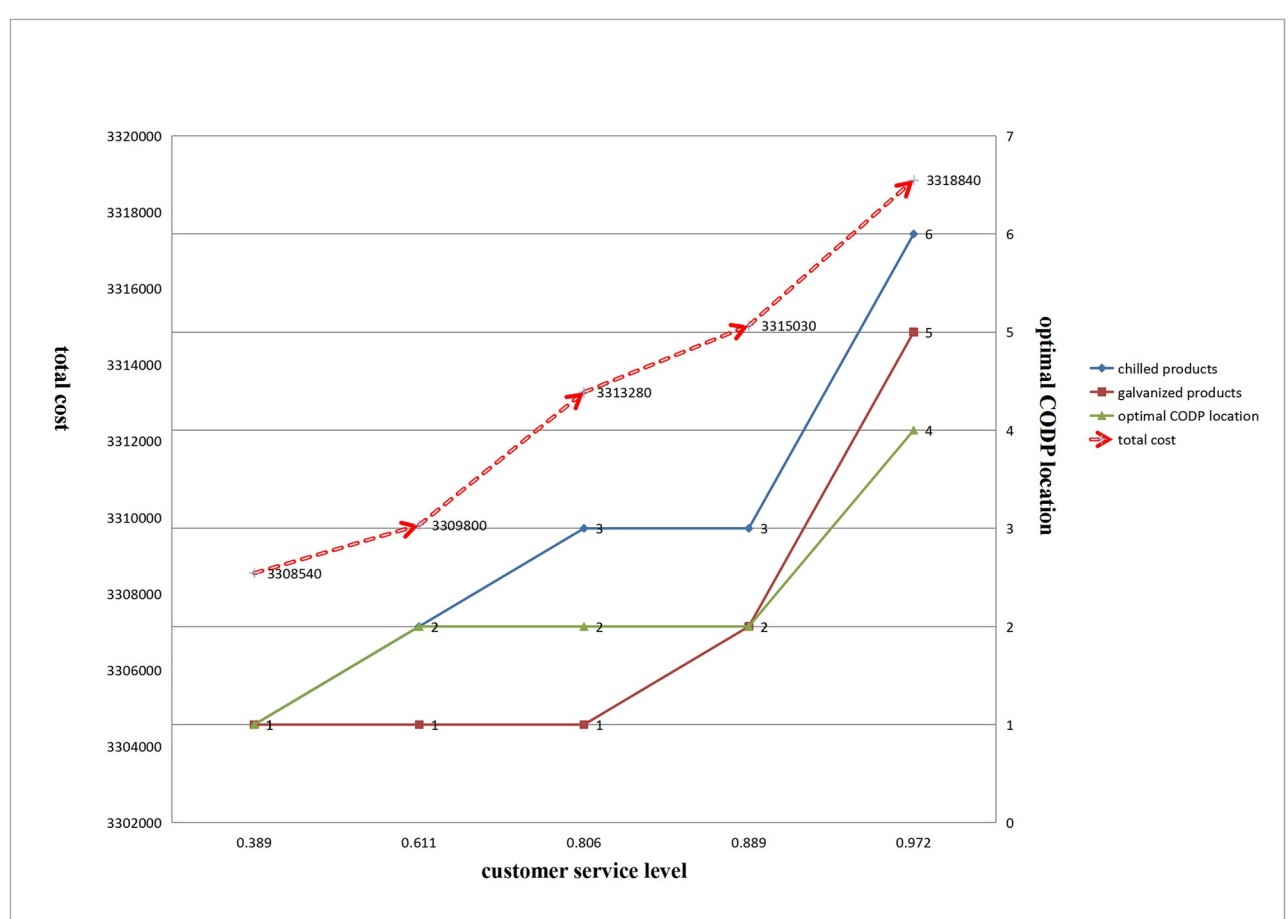

**Fig 9. Impact of changes in customer service level on total cost and the optimal location of CODP.**

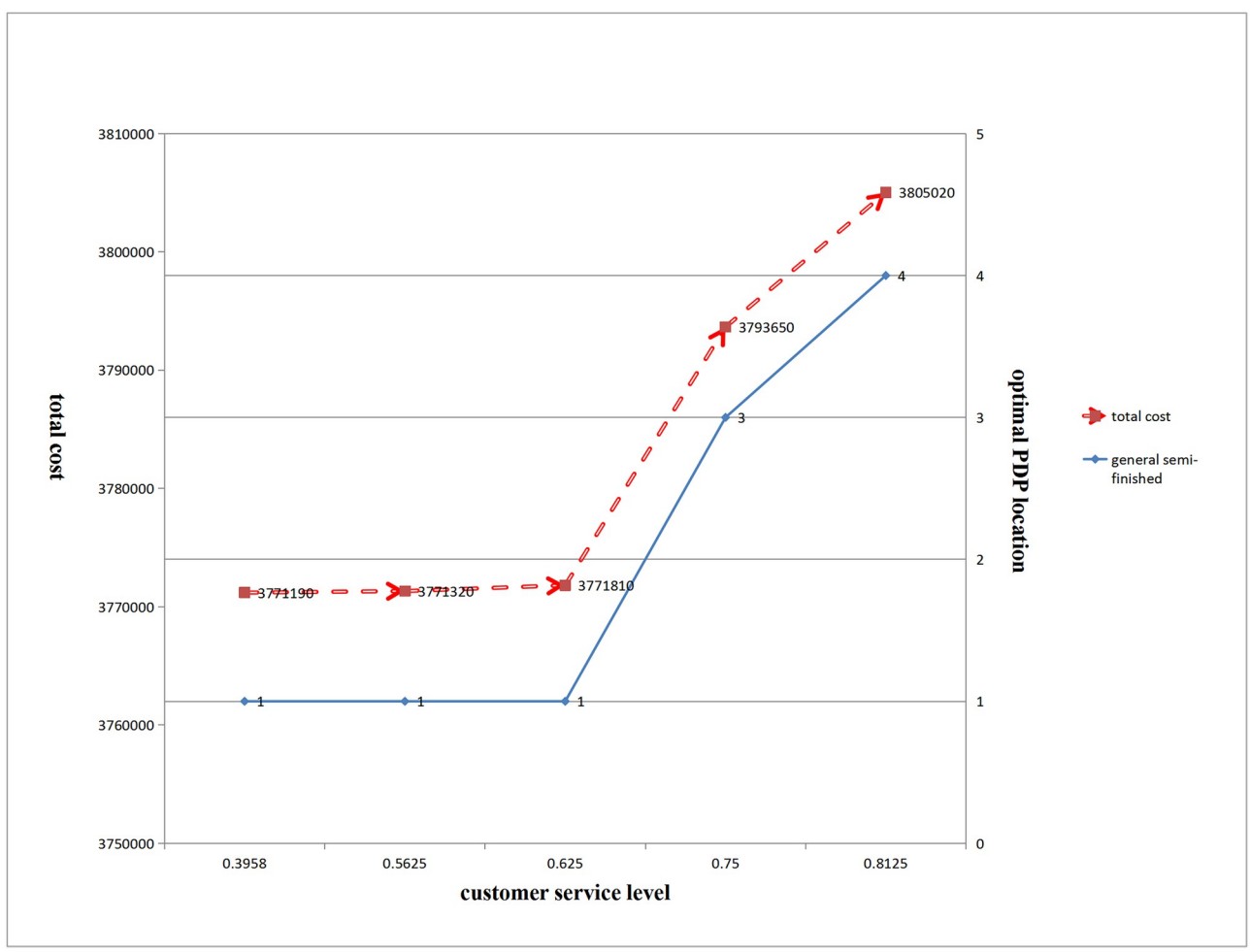

**Fig 10. Impact of changes in customer service level on total cost and the optimal location of PDP.**

Fig 9 shows that with improvement in customer service level, the optimal location of CODP for the three products gradually moved from the production start end to the production end end, and the total cost also gradually increased.

(2) Impact of changes in customer service level on the optimal PDP location and total cost

The optimal location and total cost of the general semi-finished PDP for hot-rolled, galvanized, and chilled products are shown in Fig 10:

Fig 10 shows that with an improvement in customer service level, the optimal location of general semi-finished PDP for the three products gradually moved from the production start end to the production end end, and the total cost also gradually increased.

## 5.2 Influence of delay penalty coefficient

The influence of the change of the delay penalty coefficient on the optimal location of CODP, optimal location of PDP, and total cost was analyzed. The specific analysis results are as follows.

(1) Influence of delay penalty coefficient change on CODP optimal location and total cost

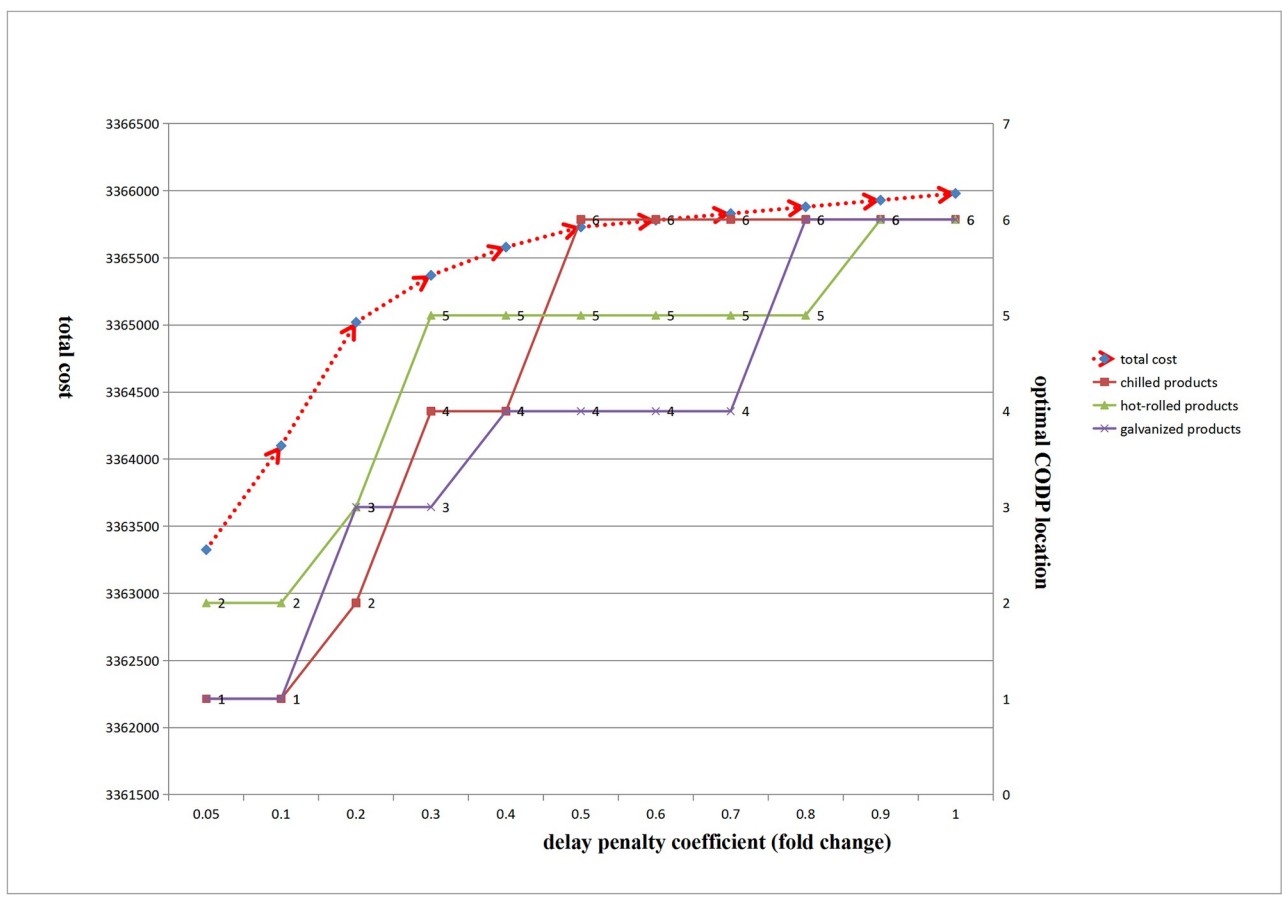

**Fig 11. Influence of the change of delay penalty coefficient on total cost and the optimal location of CODP.**

Fig 11 shows that with an increase in the delay penalty coefficient, the optimal CODP location of the three products gradually moved to the right, and the total cost gradually increased.

(2) Influence of delay penalty coefficient change on PDP optimal location and total cost

Fig 12 shows that with an increase in the delay penalty coefficient, the PDP optimal location of the general semi-finished products of the three products gradually moved to the right, and the overall gradually increased.

## 5.3 Influence of production cost variances

Numerical analysis indicated that changing the production cost of each candidate location had no effect on the change of the optimal locations of CODP and PDP. However, changing the difference in production cost between each candidate location by multiple has an effect on the optimal locations of CODP and PDP and total cost. The specific analysis results are as follows.

(1) Influence of changes in production cost differences on the optimal location of CODP and total cost

Fig 13 shows that with an increase in the production cost difference between the CODP candidate locations, when the cost difference was small, the optimal CODP location was fixed

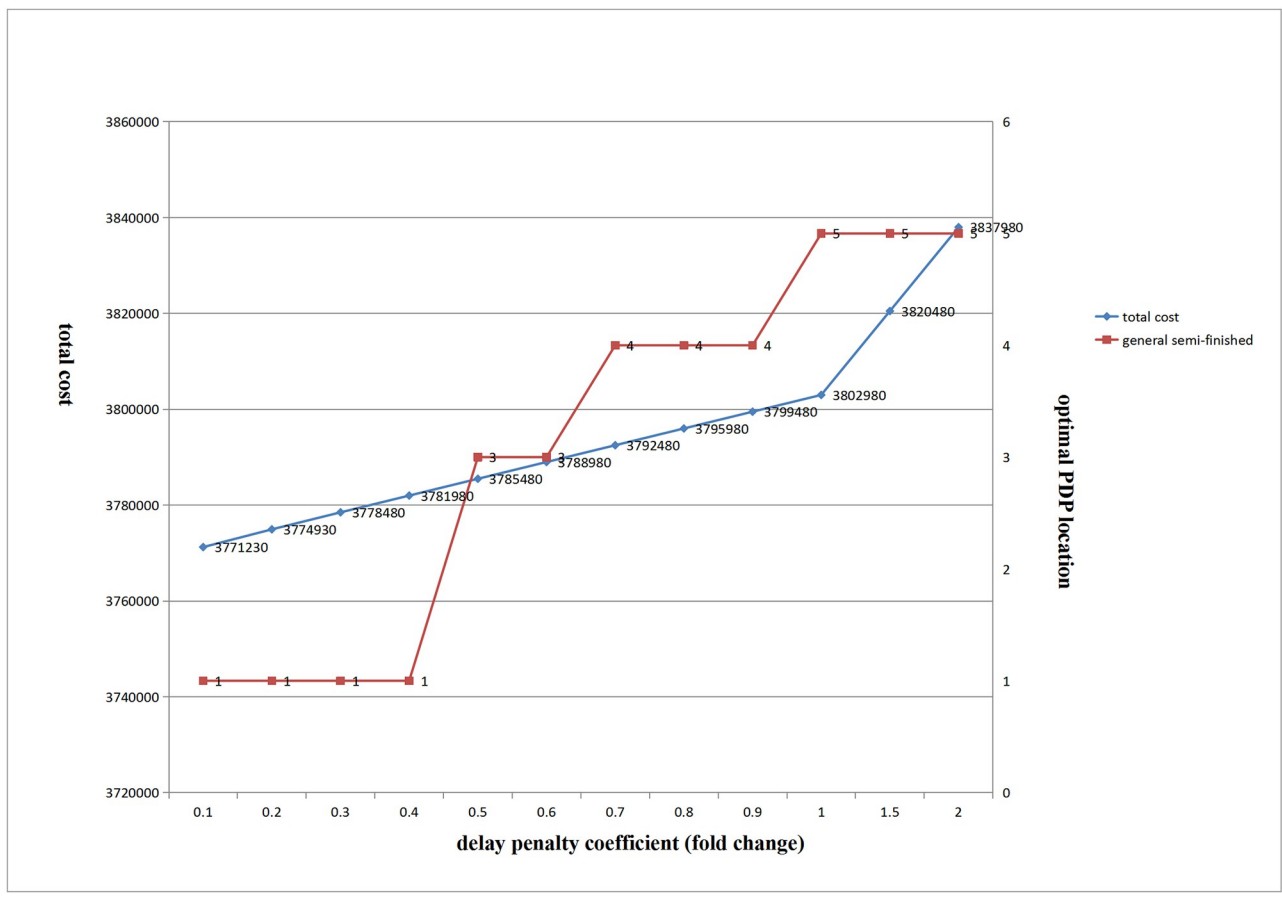

**Fig 12. Influence of delay penalty coefficient change on total cost and the optimal location of PDP.**

at the beginning of production, and total cost remained unchanged. As cost difference further increased, the optimal location of CODP suddenly changed to location 6, and the total cost gradually decreased.

(2) Influence of changes in production cost differences on the optimal location of PDP and total cost

Fig 14 shows that with an increase in the production cost difference between the PDP candidate locations, when the cost difference was small, the optimal PDP location was fixed at the beginning of production, and total cost remained unchanged. As cost difference further increased, the optimal location of PDP abruptly changed to location 5, and total cost gradually decreased.

## 5.4 Influence of early delivery benefit coefficient

This paper changed the delivery yield by multiples and analyzed its impact on the CODP and PDP optimal locations and total cost. The specific analysis results are as follows.

(1) Influence of changes in early delivery benefit on the optimal CODP location and total cost

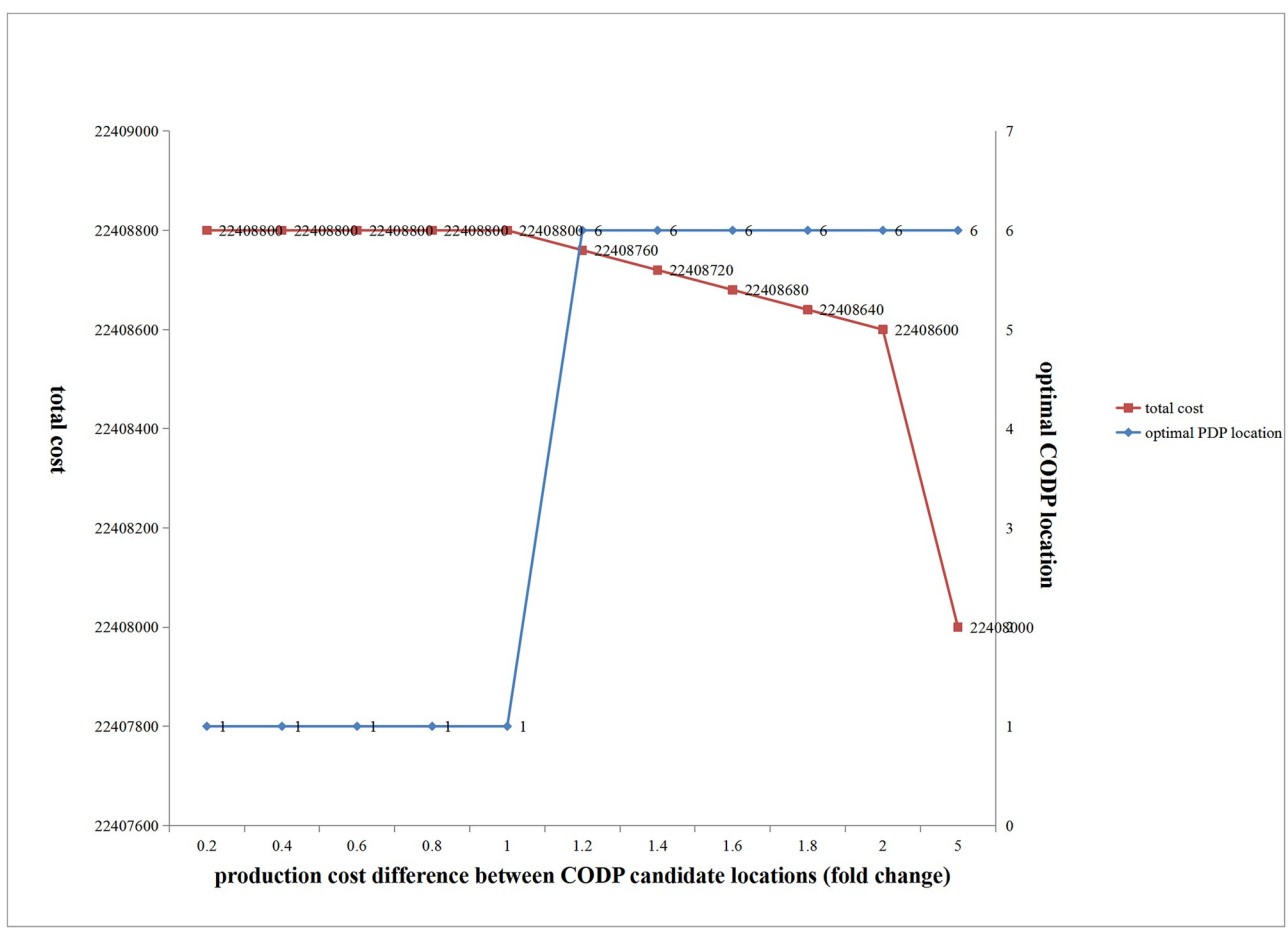

**Fig 13. Influence of production cost difference between CODP candidate locations on total cost and the optimal location of CODP.**

Fig 15 shows that with an increase in the product early delivery benefit coefficient, when the coefficient was small, the optimal location of CODP was fixed at the beginning of production, and the total cost gradually decreased. As the coefficient further increased, the optimal location of CODP abruptly changed to location 6, and total cost gradually reduced.

(2) Influence of changes in early delivery benefit on the optimal PDP location and total cost

Fig 16 shows that with an increase in the product early delivery benefit coefficient, when the coefficient was small, the optimal location of PDP was fixed at the beginning of production, and the total cost gradually decreased. As the coefficient further increased, the optimal location of PDP suddenly changed to location 5, and total cost gradually reduced.

According to the above analysis, changes in customer service level and delay penalty coefficient have a gradual and gentle impact on the delayed production control and the total cost. The change of production cost and early delivery benefit coefficient has a gradual and gentle impact on the total cost, but has an abrupt impact on delayed production control. This is because the change of production cost between the CODP and PDP candidate locations is floating, and the production cost increases first and then decreases along the production flow. In addition, the discontinuity of the influence of the early delivery benefit coefficient on the control of delayed production is because in enterprises, the customer service level is usually

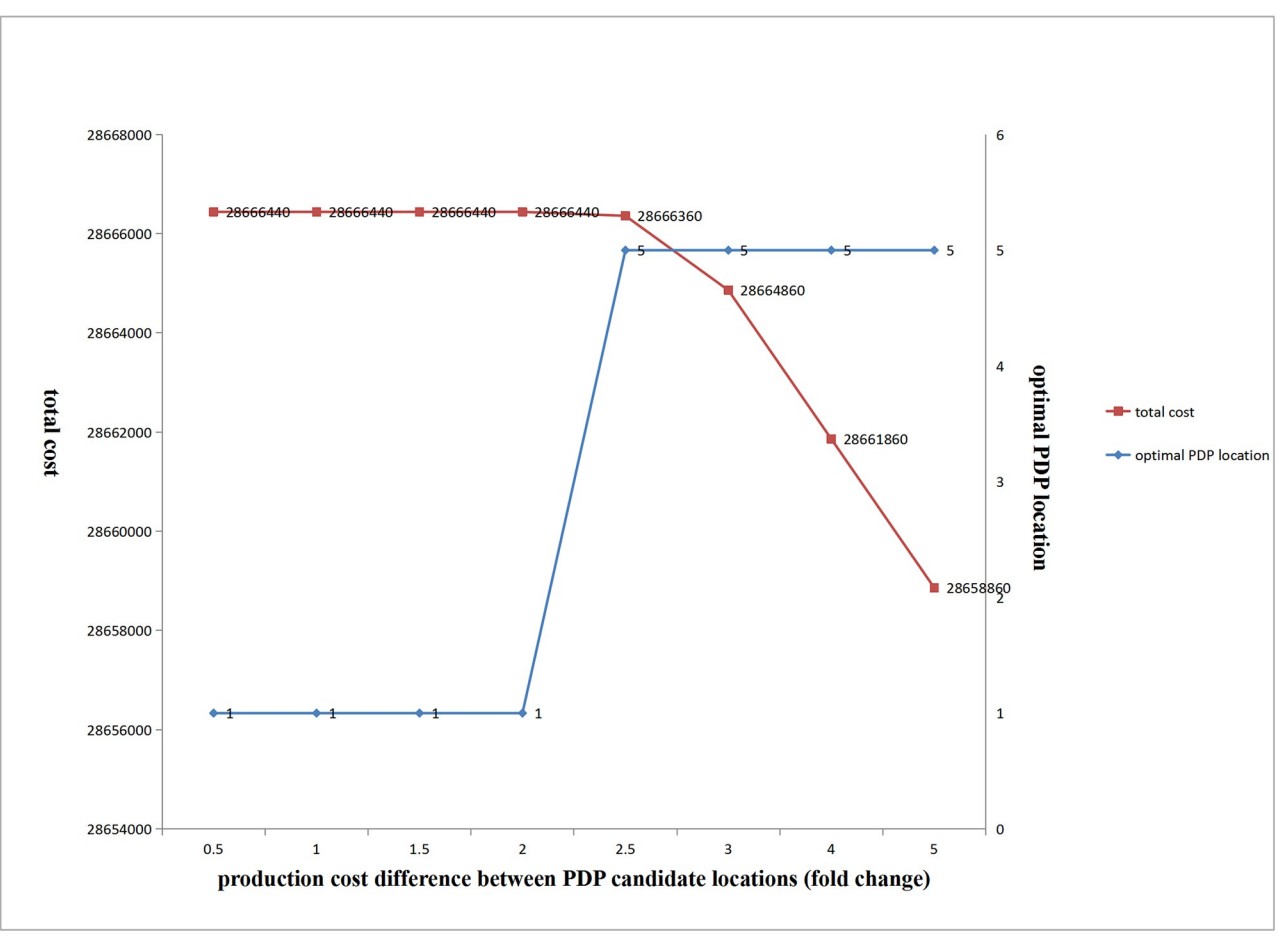

**Fig 14. Influence of production cost difference between PDP candidate locations on total cost and optimal location of PDP.**

high, most products will be delivered in advance, and the change of the early delivery benefit coefficient will affect all products produced, resulting in a large cost change from the small change of the early delivery benefit coefficient.

## 6. Conclusions

This paper used data mining method to analyze and determine the factors that strongly influence delayed production inside and outside an enterprise. Combined with the production and operation of iron and steel enterprises and the research content, several factors with a strong influence on the delayed production mode were determined, and a quantitative model of the delayed production mode was constructed thereafter. Lastly, with the help of the proposed model and actual data of the enterprise, the validity of the model was verified and key factors were analyzed. Analysis indicated that the proposed model can considerably describe the overall situation and internal influence relationship of delayed production mode. Moreover, related factors have different forms and degrees of influence on delayed production. The current study obtained the following results.

1. With an increase in customer service level or delay penalty coefficient, the optimal locations of CODP and PDP moved toward the end of production, and the total cost increased continuously. Further analysis showed that this change in delayed production aimed to meet

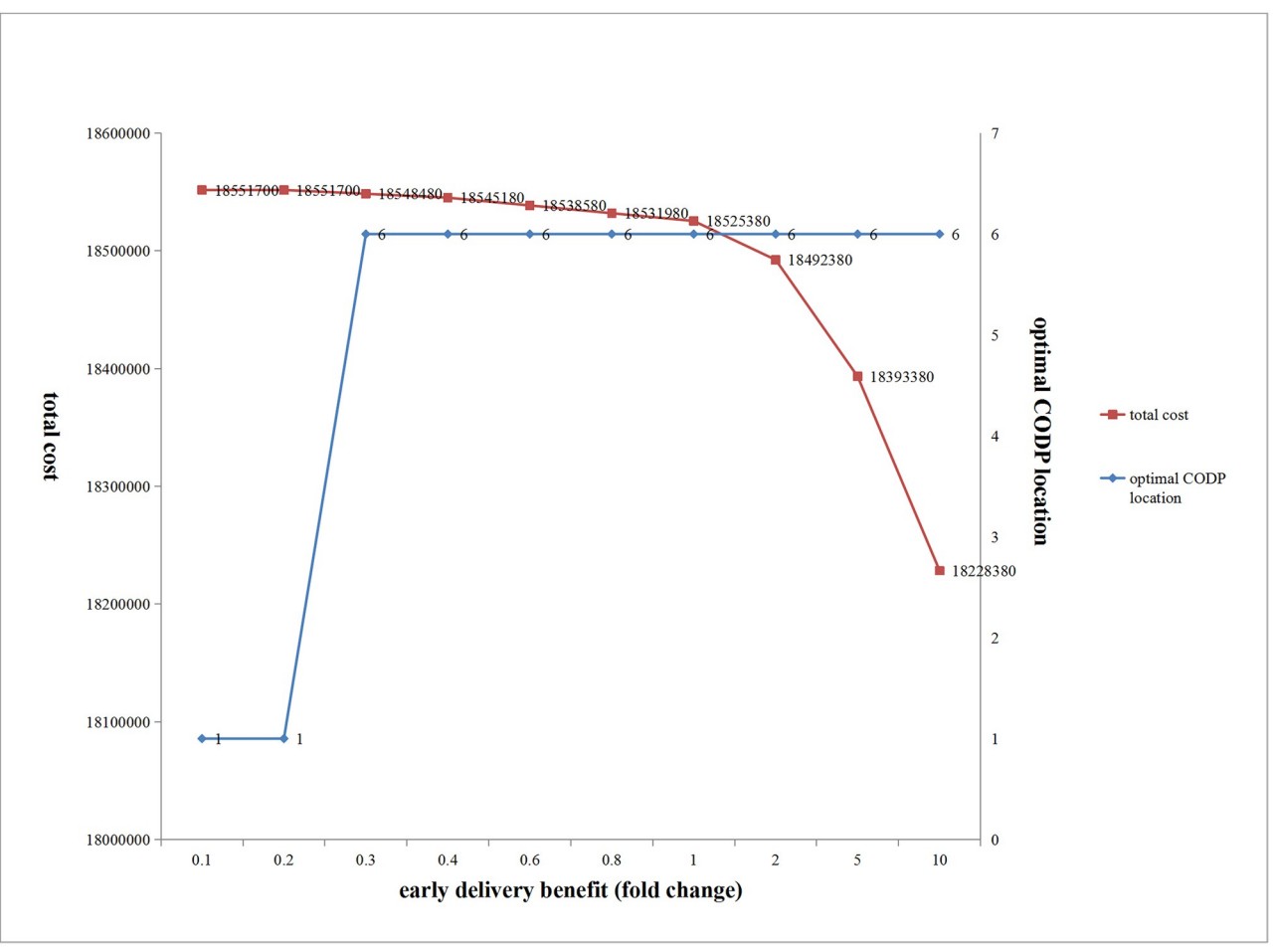

**Fig 15. Influence of the change of early delivery benefit on the optimal location and total cost of CODP.**

customer service levels and reduce losses caused by delays. Hence, semi-finished products with a high degree of processing were stored as much as possible.

2. With an increase in production cost difference and early delivery benefit coefficient, the optimal locations of CODP and PDP at the beginning of production were fixed, and the total cost changed little. The subsequent optimal locations of CODP and PDP suddenly jumped to the end of production, and the total cost began to increase. Further analysis showed that an increase in production cost difference and early delivery benefit coefficient promoted the increase of the continuity of "general production process" and "special production process" in delayed production. That is, the optimal locations of CODP and PDP were at the beginning or end of the corresponding stage. An increase in production continuity can reduce production cost and shorten production time.

This paper used data mining technology to deepen the quantitative research on delayed production mode of iron and steel enterprises. Combined with the model constructed and the experimental data in this paper, the research results of can be replicated and verified, and this research method for delayed production mode in continuous production enterprises can be reused. Its research methods and modeling ideas have quantitative practicability and can be directly applied to the delayed production process in continuous production enterprises,

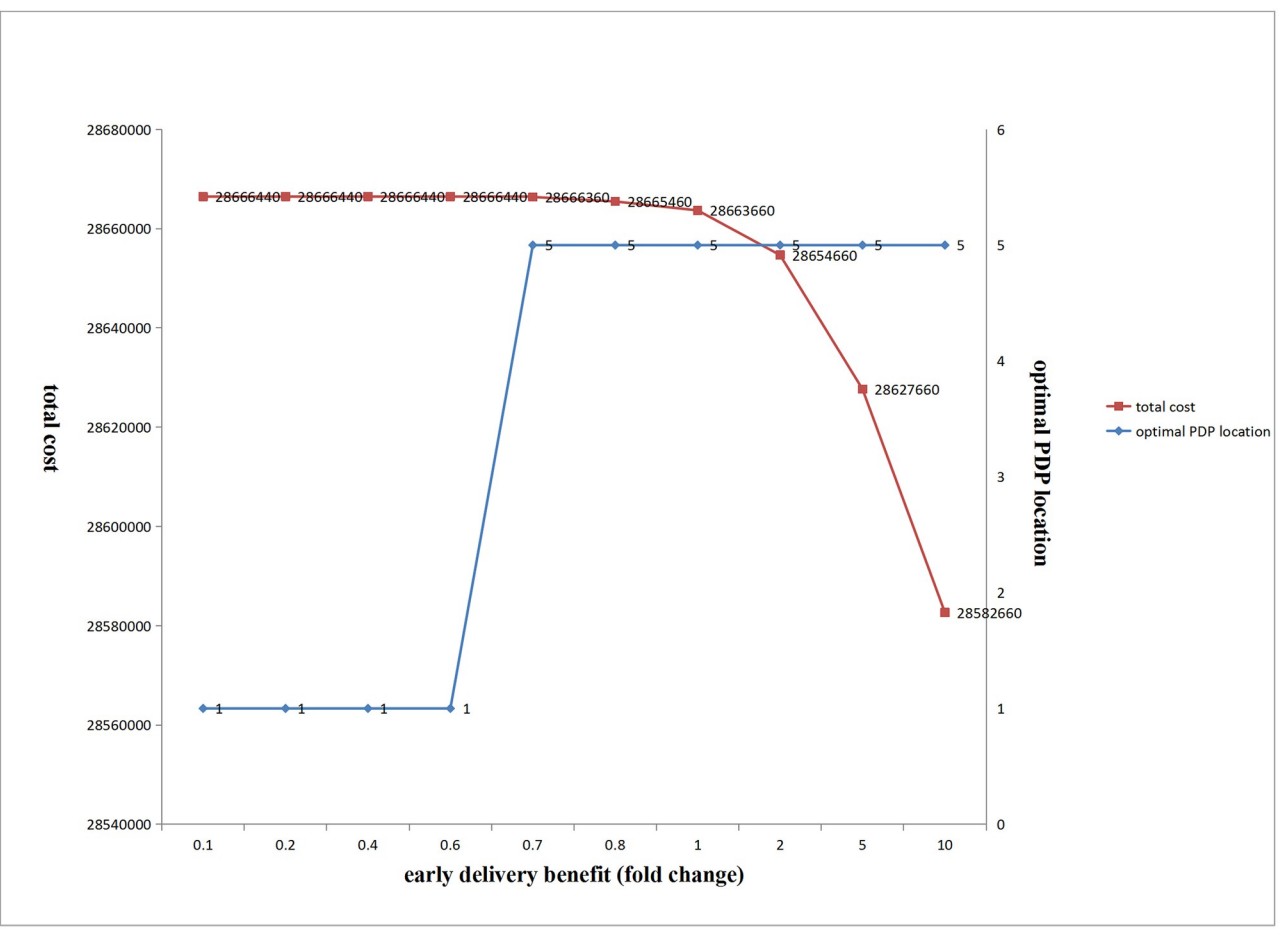

**Fig 16. Influence of the change of early delivery benefit on the total cost and the optimal location of PDP.**

including oil refineries, chemical plants, glass processing plants, food processing plants, etc. This research is to help enterprises use data mining technology to more effectively consider various internal and external factors on delayed production, and then effectively optimize the model.

This study has limitations, mainly in three aspects: First, the model does not consider the impact of multi cycle, and some continuous production enterprises may adjust the location of PDP or CODP in a short time, so it is necessary to consider the situation of multi cycle; Second, the C-LSTM artificial neural network algorithm used in this paper may not be applicable to all types of data when processing data inside and outside the enterprise, so more data mining methods should be considered; Third, the data collected in this paper is incomplete, so the results of relevant analysis may need to be further verified.

Future research can be conducted from three aspects. First, we can attempt to use a variety of data mining methods to study the influencing factors and confirm each one and to determine the factors that affect the delayed production mode more objectively. Second, this research used the heuristic algorithm in the process of solving, which may be inaccurate or biased in the analysis. Moreover, we can combine the accurate algorithm with the heuristic algorithm for comparative analysis in the later stage. Lastly, the improved quantitative model can be studied in multiple cycles to adapt to more application scenarios.

## Author Contributions

**Conceptualization:** Zhiming Shi.

**Data curation:** Zhiming Shi.

**Formal analysis:** Changxiang Lu.

**Investigation:** Zhiming Shi.

**Methodology:** Zhiming Shi, Yisong Li.

**Resources:** Zhiming Shi.

**Software:** Zhiming Shi.

**Validation:** Zhiming Shi.

**Visualization:** Zhiming Shi.

**Writing – original draft:** Zhiming Shi.

**Writing – review & editing:** Zhiming Shi, Yisong Li, Changxiang Lu.

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
