## [Decision Letter · Decision Letter 0]

28 Sep 2022

PONE-D-22-23670Optimization of delayed production mode of iron and steel enterprises based on data mining technologyPLOS ONE

Dear Dr. Shi,

Thank you for submitting your manuscript to PLOS ONE. After careful consideration, we feel that it has merit but does not fully meet PLOS ONE’s publication criteria as it currently stands. Therefore, we invite you to submit a revised version of the manuscript that addresses the points raised during the review process.

We look forward to receiving your revised manuscript.

Kind regards,

Kapil Kumar Nagwanshi, PhD

Academic Editor

PLOS ONE

Journal Requirements:

Reviewers' comments:

Reviewer's Responses to Questions

**Comments to the Author**

1. Is the manuscript technically sound, and do the data support the conclusions?

Reviewer #1: Yes

Reviewer #2: Partly

Reviewer #3: Yes

2. Has the statistical analysis been performed appropriately and rigorously? 

Reviewer #1: Yes

Reviewer #2: N/A

Reviewer #3: Yes

3. Have the authors made all data underlying the findings in their manuscript fully available?

Reviewer #1: Yes

Reviewer #2: Yes

Reviewer #3: No

4. Is the manuscript presented in an intelligible fashion and written in standard English?

Reviewer #1: Yes

Reviewer #2: Yes

Reviewer #3: Yes

5. Review Comments to the Author

Reviewer #1: 1. Poor Literature Review: Authors need to include latest and recent research articles.

2. Authors need to follow the Journal format for publication.

3. References should be in a proper format as per the journal.

4. Reference [3], Author should write the full name of journal.

5. Reference [24], Author should write the name of journal.

Reviewer #2: In this paper authors have used the convolutional neural network-long short-term memory artificial neural network algorithm(C-LSTM) in data mining technology to analyze and determine factors that have an impact on delayed production mode in the internal and external production and operation of enterprises.

Some of the suggestions are as follows:

1. Discuss the related work in detail. It will help the readers to understand the recent developments. Add papers from 2020, 2021 and 2022. Add a paragraph for the overall review analysis and outcome at the end of the literature.

2. Add numerical outcomes of results in the abstract.

3. Discuss the corelation between attributes.

4. Provide the comparison with different features.

5. The improvement should be specified properly with the reasons, justification and comparative study.

6. Include proper discussion of the results impact.

7. What are the limitations of the approach presented?

8. Image quality should be improved.

9. How the results can be replicated in the future?

10. Discuss the impact of external factors.

Reviewer #3: 1. Abstract :- Is there any significant improvement in accuracy between experiment with conventional and proposed method with respect to CODP and PDP? Please give improvement result in percentage.

2. Dataset:- Data set analysis is not cleared understandable ? What is the size of data set and where is the link from which you obtained without any restriction?

3. In figure 7, it has not been cleared the error loss by C-LSTM and Random forest with respect to the chilled ,hot-rolled and galvanized products.

4. Please provide the error loss in percentages for both proposed model and state of art models in table form for figure 7.

5.In model construction 4.2, It is not cleared how model generate the optimization results as it should have compared the existing optimization results!

6. In Working of Proposed model i.e combination of CNN and LSTM should be more precisely and apparent.

6. PLOS authors have the option to publish the peer review history of their article (what does this mean?). If published, this will include your full peer review and any attached files.

Reviewer #1: No

Reviewer #2: No

Reviewer #3: No

---

## [Author Response · Author response to Decision Letter 0]

4 Nov 2022

Please read the "Respond to Reviewers" letter for details

---

## [Decision Letter · Decision Letter 1]

23 Nov 2022

A Study on optimization of delayed production mode of iron and steel enterprises based on data mining

PONE-D-22-23670R1

Dear Dr. Shi,

We’re pleased to inform you that your manuscript has been judged scientifically suitable for publication and will be formally accepted for publication once it meets all outstanding technical requirements.

Kind regards,

Kapil Kumar Nagwanshi, PhD

Academic Editor

PLOS ONE

Additional Editor Comments (optional):

Reviewers' comments:

Reviewer's Responses to Questions

**Comments to the Author**

1. If the authors have adequately addressed your comments raised in a previous round of review and you feel that this manuscript is now acceptable for publication, you may indicate that here to bypass the “Comments to the Author” section, enter your conflict of interest statement in the “Confidential to Editor” section, and submit your "Accept" recommendation.

Reviewer #2: All comments have been addressed

Reviewer #3: All comments have been addressed

2. Is the manuscript technically sound, and do the data support the conclusions?

Reviewer #2: Yes

Reviewer #3: Yes

3. Has the statistical analysis been performed appropriately and rigorously? 

Reviewer #2: Yes

Reviewer #3: Yes

4. Have the authors made all data underlying the findings in their manuscript fully available?

Reviewer #2: Yes

Reviewer #3: Yes

5. Is the manuscript presented in an intelligible fashion and written in standard English?

Reviewer #2: Yes

Reviewer #3: Yes

6. Review Comments to the Author

Reviewer #2: (No Response)

Reviewer #3: Dear Editor,

With the due respect, Now manuscript may be accepted.

Thank you to All with best regard,

7. PLOS authors have the option to publish the peer review history of their article (what does this mean?). If published, this will include your full peer review and any attached files.

Reviewer #2: No

Reviewer #3: No

---

## [Editor Report · Acceptance letter]

6 Jan 2023

PONE-D-22-23670R1 

A Study on optimization of delayed production mode of iron and steel enterprises based on data mining 

Dear Dr. Shi:

I'm pleased to inform you that your manuscript has been deemed suitable for publication in PLOS ONE. Congratulations! Your manuscript is now with our production department. 

Kind regards, 

on behalf of

Dr. Kapil Kumar Nagwanshi 

Academic Editor

PLOS ONE